Differences in the endophytic fungal community and effective ingredients in root of three Glycyrrhiza species in Xinjiang, China

Dang Hanli
Zhang Tao
Wang Zhongke
Li Guifang
Zhao Wenqin
Lv Xinhua
Zhuang Li 3033573705@qq.com
College of Life Sciences, Shihezi University , Shihezi , Xinjiang , China
Baptista Paula
Electronic publication date: 2021 Mar 9
Publication date: 2021
Volume: 9
Electronic Location ID: e11047
Received 2020 Nov 23; Accepted 2021 Feb 10
Copyright: ©2021 Dang et al.
Copyright year: 2021
Copyright holder: Dang et al.
License: This is an open access article distributed under the terms of the Creative Commons Attribution License, which permits unrestricted use, distribution, reproduction and adaptation in any medium and for any purpose provided that it is properly attributed. For attribution, the original author(s), title, publication source (PeerJ) and either DOI or URL of the article must be cited.
License URL: https://creativecommons.org/licenses/by/4.0/

Keywords: Effective ingredients, Endophytic fungal community, Glycyrrhiza, Soil physicochemical, Root depth

Funding: National Natural Science Foundation of China NO. 41561010 and 31560177 This work was supported by the National Natural Science Foundation of China (NO. 41561010 and 31560177). The funders had no role in study design, data collection and analysis, decision to publish, or preparation of the manuscript.

==============================
Background

Endophytic fungi influence the quality and quantity of the medicinal plant’s bioactive compounds through specific fungus-host interactions. Nevertheless, due to the paucity of information, the composition of endophytic fungal communities and the mechanism by which effective ingredients regulate endophytic fungal communities in roots remains unclear.

Methods

In this study, we collected root and soil samples (depth range: 0–20, 20–40, and 40–60 cm) of three Glycyrrhiza species (Glycyrrhiza uralensis, Glycyrrhiza inflata, and Glycyrrhiza glabra). Glycyrrhizic acid and liquiritin content were determined using high-performance liquid chromatography (HPLC), and total flavonoid content was determined using ultraviolet spectrophotometry. High-throughput sequencing technology was employed to explore the composition and diversity of the endophytic fungal community in different root segments of three Glycyrrhiza species. Furthermore, soil samples were subjected to physicochemical analyses.

Results

We observed that the liquiritin content was not affected by the root depth (0–20 cm, 20–40 cm, and 40–60 cm). Still, it was significantly affected by the Glycyrrhiza species (Glycyrrhiza uralensis, Glycyrrhiza inflata, Glycyrrhiza glabra) (P < 0.05). In Glycyrrhiza root, a total of eight phyla and 140 genera were annotated so far, out of which Ascomycota and Basidiomycota phyla, and the Fusarium, Paraphoma, and Helminthosporium genera were found to be significantly dominant. Spearman correlation analysis revealed that liquiritin content was accountable for the differences in the diversity of the endophytic fungal community. Furthermore, distance-based redundancy analysis (db-RDA) showed that physicochemical properties of the soil (available potassium and ammonium nitrogen) and the root factors (liquiritin and water content) were the main contributing factors for the variations in the overall structure of the endophytic fungal community. Our results showed that the effective ingredients of Glycyrrhiza root and physicochemical properties of the soil regulated the endophytic fungal community composition and medicinal licorice diversity.

Introduction

Glycyrrhiza species are widely grown perennial herbs in arid and semi-arid regions (Zhang & Wang, 2005). Three Glycyrrhiza species stipulated in Chinese Pharmacopeia, namely dried root and rhizome of are Glycyrrhiza uralensis, Glycyrrhiza inflata, and Glycyrrhiza glabra. Its dried roots and rhizomes are widely used as herbal medicines in eastern and western countries (Rizzato et al., 2017). A wide variety of effective ingredients, such as triterpene saponins, polysaccharides, and flavonoids (TianshuiNiu et al., 2009), are extracted from the roots of Glycyrrhiza (Wang et al., 2015). Glycyrrhizic acid, the chief triterpene saponin (Li-Ping, Cui-Ai & Hong-Yan, 2010), had demonstrated anti-inflammatory (Schröfelbauer et al., 2009), antiviral, and immunoregulatory effects (Baba & Shigeta, 1987; Crance et al., 2003). Liquiritin is a major component of flavonoids that mainly exerts anti-inflammatory (Yin et al., 2018), antioxidant, and antibacterial effects (Weidner et al. , 2009; Antolak, Czyzowska & Kregiel, 2016). Due to its medicinal and economic value, the medicinal licorice plant has become a major research hotspot. The majority of the studies on licorice plants are focused on improving licorice content in licorice plants and discerning their ecological characteristics.

As per the conventional view, the quality and quantity of the bioactive compounds extracted from medicinal plants are primarily influenced by the genetic background of the medicinal plants, the ecological environment of the plant, and soil nutrients (Ncube Finnie et al., 2012; Han et al., 2013). However, recent studies (Faeth, 2002; Huang et al, 2007; Shah et al., 2016; He et al, 2020) have shown that endophytic fungi substantially influences the quality and quantity of bioactive compounds in medicinal plants through specific fungus-host interactions.

Endophytes, in particular, endophytic fungi, are one of the most crucial components of the micro-ecosystems of plants (Min et al., 2016). Endophytic fungi form a symbiotic relationship with its host plant, and it inhabits and grows in different healthy tissues of the host plant, including stems (Vaz et al., 2009), leaves (Hernawati, Wiyono & Santoso, 2011), and roots (Radić et al., 2014). Endophytic fungi sequester carbohydrates and other nutrients from the host plant for its own growth (Singh & Mukerji, 2006) and, in exchange, confer multiple benefits to host plants. Endophytic fungi can promote the growth of host plants by increasing levels of growth hormones, such as gibberellin, indoleacetic acid, abscisic acid, and zeatin (Zhang et al., 1999). It also enhances the resistance of host plants to environmental stress by increasing the production of biologically bioactive compounds (Zhao et al. , 2011; Ratnaweera et al., 2015). For instance, endophytic fungi promote plant growth and abiotic stress resistance in wheat plants (Farhana et al., 2019). Besides, endophytic fungi increase the accumulation of secondary metabolites, such as paclitaxel and deoxypodophyllotoxin in the host plant (Firáková, Šturdíková & Múčková, 2007), thereby affecting the quantity and quality of bioactive compounds of medicinal plants.

Endophytic fungi have demonstrated high biodiversity and are widely distributed in a myriad of terrestrial and aquatic plants (Saikkonen et al., 1998). Endophytic fungi were isolated from multiple plants species, which includes important cash crop species (Pimentel et al., 2006), such as soybean, and medicinal plant species (Liu et al., 2017; Coutlyne et al., 2018), such as Dendrobium officinale and Sceletium tortuosum. However, it is noteworthy that the rapid development of high-throughput sequencing technology and bioinformatics has enabled the identification of a plethora of novel fungal species (Taylor et al., 2014). Previous studies based on high-throughput sequencing technology have speculated that there are around 5.1 million fungal species, the majority of which are symbionts (Blackwell, 2011). Currently, only a small proportion of endophytic fungi could be isolated and identified, and the majority of the medicinal plant’ endophytic fungi could not be cultured on routinely used media (Kivlin et al., 2017). Therefore, it is indispensable to detect the endophytic fungal community in medicinal plants by adopting non-conventional culture methods. Modern molecular technology, specifically Illumina high-throughput sequencing technology, had comprehensively and accurately detected the diversity of endophytic fungal communities in medicinal plants (Berg, 2009; Kathrin et al. , 2015). Next-generation sequencing, a high-throughput sequencing technique, is a more robust and accurate characterization technique for the microbial community than 18S rDNA-based non-culture methods and conventional culturing methods.

Numerous studies (Karliński & Rudawska, 2010) have shown that the host’s genetic background (genotype or species) determines the composition of endophytic fungi. Soil fertility and the ecological environment, which directly affect the content of bioactive compounds in medicinal plants, showed indirect effects on the composition and structure of the endophytic fungal community (Min et al., 2016). However, so far, there is insufficient information on the composition of endophytic fungi in the root of medicinal licorice of different genetic backgrounds (species) and soil environmental factors that affect the community structure of endophytic fungi in the root of medicinal licorice plants. Thus, in this study, we investigated the distribution and composition of endophytic fungal species of three distinct medicinal licorices at three different root depths using high-throughput sequencing and explored their relationship with effective ingredients in the root of host plants and soil’ physicochemical properties. The outcomes of this study will enhance researchers’ understanding of the environmental and host factors that influence endophytic fungi and the symbiotic relationship between endophytic fungi and medicinal plants. This study provides reference data for licorice growth for commercial and medicinal use.

Materials & Methods

Sample collection

The roots and rhizospheric soils samples (all samples were 0–20 cm, 20–40 cm, and 40–60 cm, respectively) of three Glycyrrhiza plants (Glycyrrhiza uralensis, Glycyrrhiza inflata, and Glycyrrhiza glabra) were collected during August-September, 2019 from specimens growing at three distinct sites in three different eco-regions of Xinjiang province, China. The geographical location of sampling points and soil’ physical and chemical properties are shown in Table S1. To increase the statistical significance of the study we randomly selected three healthy medicinal licorices plants from each geographical location as per the five-point sampling method, and all root samples were cut with sterile scissors. The roots of each plant were divided into three depth segments: upper (0–20 cm), middle (20–40 cm), and lower (40–60 cm). Roots of each segment were equally divided into two parts: one part was used to determine the effective ingredients of the Glycyrrhiza root samples and placed in a sterile plastic bag, and the second part of the sample was put into a sterile bag and shipped to the laboratory in ice boxes for microbial characterization. The soil and root materials from each eco-region were collected as described above.

Surface sterilization

To remove the interference of other microbes, the surface of the licorice root was disinfected and sterilized in the laboratory as described previously (Saude et al., 2008). The samples from the final rinse solution were cultured using the potato dextrose agar (PDA) plate for 72 h at 28 °C. No fungal growth was observed on the PDA media, which suggested that root samples were effectively surface-sterilized ((Cui Vinod & Gang, 2018)). All root samples were labeled and immediately placed on ice and stored at liquid nitrogen until total DNA extraction.

Physicochemical analysis of the soil

For the physicochemical analysis of rhizospheric soil, the soil samples were air-dried and sieved (2 mm mesh), and the physicochemical analysis was performed as described previously by Bao (2008). Soil pH (soil: distilled water in 1: 5 ratio) was measured using a pH meter, and soil water content (SWC) was measured by weighing. The content of organic matter (SOM) and total salt (TS) were measured by external heating with potassium dichromate and atomic absorption spectrometry, respectively. The total nitrogen (STN), total phosphorus (STP), and total potassium (STK) content were determined by the acid digestion method. 0.01 M calcium chloride extraction method was used to determine the soil nitrate-nitrogen (SNN) and soil ammonium nitrogen (SAN) levels. The available phosphorus (SAP) content was measured by the sodium bicarbonate extraction method (molybdenum-antimony colorimetry). The available potassium (SAK) content was determined by the ammonium acetate extraction method using atomic absorption spectrometry.

Determination of effective ingredients of Glycyrrhiza plant root

The Glycyrrhiza root samples were dried to constant weight, powdered using mortar and pestle, and sieved through 60-mesh. To analyze the effective ingredients, 0.2 g of sieved root powder samples were extracted using chromatographic methanol (71% concentration) in the ultrasonic bath (250 W, 40 kHz). The levels of effective ingredients, i.e., glycyrrhizic acid (GIA) and liquiritin (LI), were measured by high-performance liquid chromatography (HPLC, Agilent-1260 Infinity, USA), as described previously (Dang et al., 2020). Agilent ZORBAX SB-C18 column (150  mm × 4.6  mm, 5 µm), DAD detector, and a mobile phase (chromatographic methanol: ultra-pure water: 36% glacial acetic acid = 71:28:1; acetonitrile: 0.5% glacial acetic acid = 1:4; 5 µL injection volume; 1.0 mL −-1 elution rate) were used in the HPLC analysis. For calibration purposes, the reference materials of GIA and LI were: (CAS#1405-86-3) and LI (CAS#551-15-5), respectively, from Solarbio. The total flavonoid (GTF) content in root was measured by ultraviolet spectrophotometry at 334 nm with the liquiritin standard (CAS#551-15-5) from Solarbio as the control.

DNA extraction and library construction

Total genomic DNA was extracted from 0.5 g of root samples using the DNA Quick Plant System kit (Tiangen, China) as per the manufacturer’s protocol. The concentration and integrality of extracted DNA were detected using a NanoDrop2000 (Thermo Fisher Scientific, USA) and 1% agarose gel electrophoresis, respectively. After determining the final concentration, DNA samples were diluted to 1 ng/ µL with sterile distilled water, and each PCR product was used as template DNA. The ITS (Internal Transcribed Spacer) rDNA genes of the ITS1 region were amplified using specific primers (ITS5-1737F 5′-GGAAGTAAAAGTCGTAACAAGG-3′ and ITS2-2043R 5′-GCTGCGTTCTTCATCGATGC-3′) with barcodes (David et al., 2012). To ensure amplification efficiency and accuracy, all PCR reactions were performed with Phusion® High-Fidelity PCR Master Mix and GC Buffer (New England Biolabs). The temperature regime for PCR reactions was as follows: 95 °C/3 min, 30 cycles (95 °C/30 s, 55 °C/30 s, 72 °C/30 s), and 72 °C/5 min. PCR products were mixed with 1X loading buffer (containing SYBR green) in equidensity ratios and visualized on 2% agarose gel electrophoresis. The target sequences were purified using GeneJET™ Gel Extraction Kit (Thermo Scientific). The DNA libraries were constructed using TruSeq® DNA PCR-Free Sample Preparation Kit (Illumina, USA) as per the manufacturer’s instruction, and the quality was assessed on the Qubit® 2.0 Fluorometer (Thermo Scientific) and Agilent Bioanalyzer 2100 system. ITS sequencing was carried out with the Illumina platforms (HiSeq2500) at the Beijing Compass Biotechnology Co., Ltd. (Beijing, China).

Bioinformatics analysis and statistical analysis

Cutadapt (Liu et al., 2016) software was employed to assign the single-end reads to the respective samples based on the unique barcode, and single-end reads were truncated by cutting off the barcodes and primer sequences. Before the subsequent analysis, a total of 2,199,148 raw sequences were filtered by using Cutadapt software to remove the influence of the non-microbiota community, including chloroplast and mitochondrial sequences. Cutadapt software specific filtering conditions were used for strict quality control in order to generate high-quality clean reads. Clean reads were obtained by comparison with the reference database (Unite database) (Haas et al, 2011) using the UCHIME algorithm to detect and remove chimeric sequences.

UPARSE software (Martin, 2011) (Version 7.0.1001) was used to cluster the clean reads into the same operational taxonomic units (OTUs) with ≥ 97% similarity. The clean reads with the highest frequency were used as the representative sequence of each OTU. The classification information for each representative sequence was annotated through the Unite database based on the BLAST algorithm using QIIME software (Version 1.9.1). To decipher the phylogenetic relationship among 27 samples, MUSCLE (Version 3.8.31) software was employed for multiple sequence alignment. The OTU abundance information was normalized by the standard sequence number corresponding to the minimum sequence sample (54,262 reads for sample D.2.1).

Figure 1 Effect of main effect plant species on the effective ingredients of licorice roots.

Ordinate is the content of liquiritin (A), glycyrrhizic acid (B) and total flavonoid (C); abscissa is the group name: Gi, Gg and Gu: Glycyrrhiza inflata, Glycyrrhiza glabra and Glycyrrhiza uralensis, and different letters indicated significance test (p < 0.05). Bars (mean with standard error) with different lowercase letters indicated significant difference (P < 0.05) assessed by one-way analysis of variance followed by Bonferroni’s test for multiple comparisons.

Alpha diversity analysis based on output normalized data were used to study the complexity of species diversity in a sample using six indices (observed-species, Shannon, Simpson, Chao1, ACE, and good-coverage) (Li et al, 2013). All indices in the samples were calculated with QIIME (Version 1.7.0) and displayed with R software (Version 3.6.1).

To evaluate differences in sample species complexity, the beta diversity analysis of output-normalized data was used, which was based on weighted Unifrac and calculated using QIIME software. The Un-weighted Pair-group Method with Arithmetic Mean (UPGMA) clustering analysis was conducted by QIIME software (Version 1.7.0). In addition, R software (Version 3.6.1) was also used for rarefaction curve generation, Wilcoxon rank-sum test, Metastat statistical test, Spearman correlation analysis of heat maps, and Distance-based Redundancy Analysis (db-RDA). Pearson correlation analysis (Pearson coefficient, r) was performed for the effective ingredients and the physicochemical properties of the soil, with the significance level set to 0.05. ANOVA was performed with SPSS (Version19.0) (IBM Inc., Armonk, USA) and displayed with GraphPad Prism 5. The statistically significant differences were determined by ANOVA, followed by Bonferroni’s statistical test for multiple comparisons, and the significance level was set to 0.05.

Results

Differences in levels of effective ingredients in Glycyrrhiza roots

The effective ingredients of Glycyrrhiza roots and physicochemical properties of soil are presented in Table S2. The results of two-way ANOVA showed that the levels of the effective ingredients, i.e., glycyrrhizic acid (GIA), liquiritin (LI), and total flavonoid (GTF), were not significantly affected by the interaction between root depth (0–20 cm, 20–40 cm, and 40–60 cm) and plant species (Glycyrrhiza uralensis, Glycyrrhiza inflata, and Glycyrrhiza glabra) (P > 0.05) (Table S3). However, the content of LI was significantly affected by the main effect plant species (P < 0.05) (Table S3 and Fig. 1). As shown in Fig. 1, LI content in Glycyrrhiza uralensis (Gu) root was significantly higher than Glycyrrhiza inflata (Gi) (P < 0.05) and Glycyrrhiza glabra (Gg) roots (P < 0.05) (Fig. 1A).

As per the Pearson correlation analysis, levels of effective ingredients were significantly correlated with the physicochemical properties of the soil (Table 1). GIA content in Glycyrrhiza level was significantly and positively correlated to the available potassium (SAK) and water content of the soil (SWC) (r > 0; P < 0.05); however, LI level in root was significantly and negatively correlated to SAK and total salt (TS) content of the soil (r < 0; P < 0.05).

Table 1 Pearson correlation coefficient of the content of bioactive compounds with soil physicochemical properties.

	GlA	GTF	LI	SOM	STN	STP	STK	SNN	SAN	SAP	SAK	TS	PH	SWC	
GlA	1.000	0.419*	0.294	−0.146	0.121	0.158	−0.107	−0.144	−0.337	−0.121	0.463*	−0.069	0.263	0.609**	
GTF	0.419*	1.000	0.172	−0.142	0.132	0.345	−0.121	0.048	−0.008	0.034	0.151	−0.217	0.000	0.160	
LI	0.294	0.172	1.000	−0.031	0.183	0.251	−0.294	0.070	0.239	−0.216	−0.415*	−0.403*	0.058	−0.183	
SOM	−0.146	−0.142	−0.031	1.000	0.274	−0.527**	0.238	0.532**	0.248	0.400*	−0.176	0.136	−0.229	−0.327	
STN	0.121	0.132	0.183	0.274	1.000	0.455*	−0.249	0.416*	0.333	0.415*	−0.022	0.166	−0.236	−0.300	
STP	0.158	0.345	0.251	−0.527**	0.455*	1.000	−0.465*	−0.198	0.245	−0.045	−0.034	−0.119	0.090	−0.033	
STK	−0.107	−0.121	−0.294	0.238	−0.249	−0.465*	1.000	0.167	−0.326	0.211	0.247	0.182	−0.020	0.156	
SNN	−0.144	0.048	0.070	0.532**	0.416*	−0.198	0.167	1.000	0.267	0.736**	−0.011	0.489**	−0.284	−0.411*	
SAN	−0.337	−0.008	0.239	0.248	0.333	0.245	−0.326	0.267	1.000	0.076	−0.566**	−0.066	−0.315	−0.641**	
SAP	−0.121	0.034	−0.216	0.400*	0.415*	−0.045	0.211	0.736**	0.076	1.000	0.364	0.723**	−0.253	−0.172	
SAK	0.463*	0.151	−0.415*	−0.176	−0.022	−0.034	0.247	−0.011	−0.566**	0.364	1.000	0.556**	0.063	0.750**	
TS	−0.069	−0.217	−0.403*	0.136	0.166	−0.119	0.182	0.489**	−0.066	0.723**	0.556**	1.000	−0.062	0.238	
PH	0.263	0.000	0.058	−0.229	−0.236	0.090	−0.020	−0.284	−0.315	−0.253	0.063	−0.062	1.000	0.446*	
SWC	0.609**	0.160	−0.183	−0.327	−0.300	−0.033	0.156	−0.411*	−0.641**	−0.172	0.750**	0.238	0.446*	1.000	
Notes.

Description: the values are Pearson’s correlation coefficients (significance level was set at 0.05). The correlation coefficient r of Pearson is between −1 and 1, r < 0 means negative correlation, r > 0 means positive correlation, r = 0 means no linear correlation.

** P < 0.01.

* P < 0.05.

Abbreviations GlA glycyrrhizic acid

GTF total flavonoid

LI liquiritin

SOM soil organic matter

STN soil total nitrogen

STP soil total phosphorus

STK soil total potassium

SNN soil nitrate nitrogen

SAN soil ammonium nitrogen

SAP soil available phosphorus

SAK soil available potassium

TS total salt

PH soil pH

SWC soil water content

Sequencing of Glycrrhiza root’s endophytic fungi

A total of 2,118,633 effective sequences were identified by sequencing the root samples from three Glycyrrhiza (Glycyrrhiza uralensis, Glycyrrhiza glabra, and Glycyrrhiza inflata) species using Illumina HiSeq sequencing and after filtering out low-quality and short sequence reads. The sequencing results of each sample are listed in Table S4. The effective sequences were clustered into OTUs with 97% identity, and a total of 1,063 OTUs were obtained. Out of the total effective sequences, 91.53% were assigned to the kingdom level, 59.27% to the phylum level, 54.37% to the class level, 53.72% to the order level, 46.19% to the family level, 38.01% to the genus level, and 23.52% to the species level (Fig. S1A). The rarefaction curves showed that the number of OTU in each sample increased gradually with the sequence quantity, which validated that the sequencing data was adequate for the analysis (Fig. S1B).

Differences in alpha diversity

The alpha diversity index of each group is demonstrated in Table S5. Alpha diversity indices, Shannon and Chao1, deciphered the diversity and richness of microbial communities in Glycyrrhiza root samples. A higher index value denotes higher species diversity and distribution. The Shannon index of the Gu1 (4.910) sample was the highest, and that of the Gi1 (3.393) sample was the lowest. Moreover, the Gi1 root sample showed the lowest Chao1 (238.678) and ACE (253.105) values, while the Gi3 root sample showed the highest Chao1 (356.317) and ACE (355.694) values. As per Wilcoxon rank-sum analysis, the Shannon index showed significantly different distribution between Gu and Gi samples, especially at 0–20 cm at the root depth (Fig. 2A). The Shannon index in the Gu1 root sample was significantly higher than the Gi1 root sample (P < 0.05). Furthermore, the Chao1 index in sample Gi increased gradually with decreasing root depths, and as per the Wilcoxon rank-sum test, sample Gi’Chao1 index value was significantly affected by root depth (Fig. 2B). Specifically, the Chao1 index value of Gi3 sample was significantly higher than the Gi1 sample (P < 0.01), and that of the Gi2 root sample was significantly higher than the Gi1 root sample (P < 0.05).

Figure 2 The significance test of the differences of Alpha Diversity.

Ordinates are Shannon index (A) and Chao1 index (B), respectively. Abscissa is the group name: Gi, Gg and Gu: Glycyrrhiza inflata, Glycyrrhiza glabra and Glycyrrhiza uralensis; 1, 2 and 3: root depth 0–20 cm, 20–40 cm, and 40–60 cm, respectively. The mark * is significance test p < 0.05.

Differences in beta diversity

To evaluate differences in microbial community composition among Glycyrrhiza root samples, beta diversity analysis was performed. The Unweighted Pair-Group Method with Arithmetic (UPGMA) cluster analysis was performed to discern similarity in the composition of endophytic fungal community among different Glycyrrhiza root samples. The UPGMA clustering results were integrated with the relative abundance of species at the phyla and taxon levels in each group. The UPGMA cluster tree outcomes based on Weighted Unifrac distances showed that Gg3, Gg2, Gg1, Gu1 samples and Gu3, Gi2, Gu2, Gi1 samples were clustered together (Fig. 3A). Meanwhile, to discern the difference in the beta diversity between different groups of samples, a Wilcoxon rank-sum test based on Weighted Unifrac distances was constructed (Fig. 3B). The outcomes of this test showed that there were significant differences in beta diversity between the Gu and Gi group of samples, which was consistent with the UPGMA cluster tree. Specifically, there were significant differences in beta diversity between Gi1 and Gi2 samples (P < 0.05), Gi3 and Gu3 samples (P < 0.05), and Gi1 and Gu1 samples (P < 0.01) (Fig. 3B). It indicated that endophytic fungal community composition differed significantly between different species and different root depths in medicinal licorice plants.

Figure 3 Unweighted Pair-Group Method with Arithmetic (UPGMA) clustering tree base on the weighted unifrac distance (A) and the significance test of the differences of Beta Diversity (B).

(A) The UPGMA cluster tree structure, and the distribution of relative abundance of each sample at the phylum level; (B) Ordinate is the Beta diversity; Abscissa is the group name: Gi, Gg and Gu: Glycyrrhiza inflata, Glycyrrhiza glabra and Glycyrrhiza uralensis; 1, 2 and 3: root depth 0–20 cm, 20–40 cm, and 40–60 cm, respectively. The mark * is significance test p < 0.05.

Differences in endophytic fungal community composition in medicinal licorice

Based on OTU sequences and the Unite database, total sequences were annotated into 8 phyla, 23 classes, 53 orders, 102 families, 140 genera, and 141 species. The most abundant endophytic fungal phyla in all the nine groups are enumerated in Fig. 4A. Ascomycota phyla were found to be the most dominant phyla among all the samples, accounting for 91.821%, 60.558%, 39.956%, 79.651%, 62.305%, 54.241%, 82.176%, 81.928%, and 80.290% of the total number of species in Gi1, Gi2, Gi3, Gg1, Gg2, Gg3, Gu1, Gu2, and Gu3 samples, respectively. In addition, Basidiomycota accounted for 21.348%, 28.440%, 10.631%, 12.523%, 6.749%, and 5.110% of relative abundance in Gi2, Gi3, Gg2, Gg3, Gu2, and Gu3 samples, respectively. The relative abundance of Ascomycota phyla decreased with increasing root depth. To determine the differences at the phylum level in different groups of root samples, a MetaStat statistical test based on species abundance was conducted. As per the outcomes of this test, the relative abundance of Ascomycota in sample Gi showed significant differences in distribution at different root depths (Fig. 4B). Specifically speaking, the relative abundance of Ascomycota in the Gi1 sample (91.821%) was significantly higher than the Gi3 sample (39.956%) (Fig. 4B).

Figure 4 Histograms of relative abundance of the top 10 endophytic fungi at the phyla (A) level of taxonomy and difference analysis at the Phylum classification level (B), Histograms of relative abundance of the top 10 endophytic fungi at the genera (C) level.

Ordinate is the relative abundance; others refers to are sequences with less or not be annotated. Abscissa is the group name: Gi, Gg and Gu: Glycyrrhiza inflata, Glycyrrhiza glabra and Glycyrrhiza uralensis; 1, 2 and 3: root depth 0–20 cm, 20–40 cm, and 40–60 cm, respectively. ** P < 0.01.

At genus level, the top 10 dominant fungal genera based on relative abundance in each group (Fig. 4C) were Fusarium (Gi1: 27.907%, Gg1: 23.944%, Gg2: 31.071%, Gg3: 25.381%, Gu1: 19.253%, Gu3: 18.215%), Paraphoma (Gi1: 27.738%, Gi3: 23.937%, Gu3: 13.980%), Helminthosporium (Gi1: 26.567%, Gg1: 25.124%, Gu1: 8.224%, Gu2: 17.408%), Sarocladium (Gi2: 3.326%, Gg1: 16.547%, Gg2: 17.243%, Gg3: 21.897%, Gu1: 4.218%), Cladosporium (Gi2: 6.446%, Gi3: 2.721%, Gu3: 15.174%). Furthermore, Cadophora (13.200%) and Psathyrell a (10.917%) were found to be the most dominant genera in Gi2 sample, Tomentella (14.472%) in Gi3 sample, and Conocybe (12.068%) in Gg3 sample (Fig. 4C).

At the same time, details of the composition of the top 10 dominant fungi at other classification levels (Class, Older, Family and Species) were listed in Table S6. Specifically speaking, Sordariomycetes, Dothideomycetes, Agaricomycetes were found to be the dominant class; Hypocreales, Pleosporales, Thelephorales dominant order; Nectriaceae, Phaeosphaeriaceae, Massarinaceae dominant family; Fusarium-solani, Paraphoma-radicina, Sarocladium-kiliense dominant species.

Correlation of effective ingredients with physicochemical properties of soil and endophytic fungal community in the Glycyrrhiza roots

Spearman correlation analysis showed that the LI content was significantly and positively correlated with the alpha diversity index (r > 0, P < 0.05) (Fig. 5). Besides, LI content showed a highly significant and positive correlation with the Shannon index, Simpson index, and Chao1 index (P < 0.05). It indicated that the LI content led to the differences in the diversity of the endophytic fungal community in medical licorice roots in this study.

Figure 5 Heatmaps of Spearman correlation analysis.

Ordinate is the information of environmental factors, and abscissa is the information of alpha diversity indexes. The correlation coefficient r of Spearman is between −1 and 1, r < 0 is negative correlation, r > 0 is positive correlation, and the mark * is significance test (p < 0.05). Abbreviations: SOM, soil organic matter; STN, soil total nitrogen; STP, soil total phosphorus; STK, soil total potassium; SNN, soil nitrate nitrogen; SAN, soil ammonium nitrogen; SAP, soil available phosphorus; SAK, soil available potassium; TS, total salt; PH, soil pH; SWC, soil water content; RWC, root water content; GlA, glycyrrhizic acid; GTF, total flavonoid; LI, liquiritin.

Distance-based redundancy analysis (db-RDA) based on the Bray–Curtis distance showed that the effective ingredients and physicochemical properties of the soil had significant effects on the differences in the endophytic fungal community (Fig. 6). The differential distribution of endophytic fungal community was restricted primarily to the first and second ordination axes and explained 16.23%, 13.89% of the total variability, respectively (Fig. 6). Out of all the soil’ environmental factors, SAK content affected the differences of the endophytic fungal community most significantly (r2 = 0.329, P < 0.01), followed by SAN (P < 0.05). Among the root factors, RWC most significantly affected the difference of endophytic fungal communities (r2 = 0.247, P < 0.05), followed by LI content (P < 0.05) (Fig. 6, Table S7). As per the outcomes of the db-RDA analysis, the SAN, SAK, RWC, and LI content were the major factors contributing to the variations in the overall structure of the endophytic fungal community in the roots of the medicinal plants in the current study.

Figure 6 Distance-based redundancy analysis (db-RDA) for all groups.

Environmental factors are generally represented by arrows. The length of the arrow line represents the degree of correlation between a certain environmental factor and community and species distribution, and the longer the arrow, the greater the correlation. When the angle between the environmental factors is acute, it means that there is a positive correlation between the two environmental factors, while when the angle is obtuse, there is a negative correlation. Abbreviations: SOM, soil organic matter; STN, soil total nitrogen; STP, soil total phosphorus; STK, soil total potassium; SNN, soil nitrate nitrogen; SAN, soil ammonium nitrogen; SAP, soil available phosphorus; SAK, soil available potassium; TS, total salt; PH, soil pH; SWC, soil water content; RWC, root water content; GlA, glycyrrhizic acid; GTF, total flavonoid; LI, liquiritin. Gi, Gg and Gu: Glycyrrhiza inflata, Glycyrrhiza glabra and Glycyrrhiza uralensis; 1, 2 and 3: root depth 0–20 cm, 20–40 cm, and 40–60 cm, respectively.

Discussion

In the current study, the composition and diversity of endophytic fungal communities at different root depth range (0–20 cm, 20–40 cm, and 40–60 cm) of three Glycyrrhiza species (Glycyrrhiza uralensis, Glycyrrhiza glabra, and Glycyrrhiza infl ata) were investigated using high-throughput sequencing technology. Thus, a highly accurate and substantial amount of data was procured than previous studies based on conventional technology (Tie et al., 2010; Jiang-Tao et al., 2013; Palak et al., 2019). As per the alpha and beta diversity analysis of endophytic fungal community, fungal communities between Glycyrrhiza uralensis and Glycyrrhiza inflata showed significant differences at different root depths (0–20 cm, 20–40 cm, and 40–60 cm) (Figs. 2 and 3). It indicated that the host plant’s genotype and ecological region contributed to the differences in endophytic fungal communities. Numerous studies (Saikkonen et al., 2004) have demonstrated that the adaptation of the endophytic fungal community primarily relies on adapting host plants to the ecological environment. It implies that host plants substantially influence the colonization and distribution of the endophytic fungal communities. The interaction between fungus and host plant is often considered dynamic where orientation is determined by subtle differences in the expression of fungal genes in response to the host or, conversely, by the host’s recognition and response fungus. Thus, slight genetic differences in the two genomes control the symbiosis (Moricca & Ragazzi, 2008).

Furthermore, the current study also showed that root depth significantly affected the richness and composition of the endophytic fungal community (Figs. 2 and 3). It indicated that ecologically different fungi might represent certain ecological regions (root depth). It might be a crucial factor that should be taken into account while inoculating endophytic fungi into the host plants. We speculated that this could be correlated to root respiration and soil C content. Root respiration, which accounts for 60% of total soil respiration, regulates the metabolism of roots and soil microbes and is considered a significant contributor to the terrestrial carbon budget (Pregitzer et al., 1998). Also, the C content in unstable soil varies significantly at different soil depths (de Graaff et al., 2014). Moreover, (Fierer, Schimel & Holden, 2003) demonstrated that the vertical distribution of the specific microbial species was correlated mainly to decreased carbon availability with increasing soil depth.

In this study, we employed high-throughput sequencing to determine the composition of endophytic fungal communities at different taxonomic levels (phylum, class, older, family, genus, and species) (Figs. 4A, 4C, and Table S6). The results indicated that 27 samples of medicinal licorice roots contained a peculiar microbiome. For example, Ascomycota was the dominant phylum in all samples, followed by Basidiomycota, which was in line with previous studies (Stephenson, Tsui & Rollins, 2013; Tan et al., 2018). The phylum Ascomycota, the largest phylum of fungi, entails a highly diverse population and plays a vital role in genetics (Wallen & Perlin, 2018), ecology (Belnap & Lange, 2005), and phylogeny (López-Giráldez et al., 2009). For instance, Ascomycota produces large numbers of spores through both asexual and sexual reproduction. To disperse ascospores, asci act as small water cannon and sprays spores into the air. Spores are also spreads multiple phytopathogenic and saprophytic fungi (Trail, 2010). Most of the members of Ascomycota are saprophylaxis and plays an important role in organic matter decomposition in the soil. In this process, the dominant fungal community assimilates root exudates for organic matter degradation (Mylène et al., 2018). He et al. (2020) showed that inoculation of licorice plants with dark-colored septogenic endophyte (DSE), conidia, or sterile ascomycetes increased root biomass, uptake of nitrogen (N) and phosphorus (P) by roots, and concentration of glycyrrhizin and glycyrrhizic acid.

Moreover, outcomes of this study indicated that the relative abundance of Ascomycota gradually decreased with the increasing root depths (Fig. 4B), in line with the previous study by Ko (2015). Based on this data, we investigated the correlation between Ascomcycota in Glycyrhizza roots and root depth. We observed that the relative abundance of Ascomycota in Glycyrrhiza inflata differed significantly with root depth. However, the relative abundance of Ascomycota in Glycyrrhiza uralensis and Glycyrrhiza glabra did not differ significantly with root depth. It indicated that specific endophytes proliferate preferentially in certain ecological regions and play different ecological roles than other endophytes. Concisely, in addition to soil depth, the relative abundance of endophytes was also correlated to the genotype of the host plant species. These findings were in line with the study on host genotype and soil conditions on the ectomycorrhizal community of poplar clones (Karliński, Rudawska & Leski, 2013).

However, the primary limitation of the generalization of the current study’s outcomes is that three samples of Glycyrrhiza plant roots were collected from distinct geographical areas. As it is challenging to procure three distinct Glycyrrhiza species from the same habitat, to a certain extent, physicochemical properties of the soil can represent the environmental factors of Glycyrrhiza species. Therefore, in this study, we investigated the effect of the root as well as soil factors.

Numerous studies (Da-Cheng, (Da-Cheng & Pei-Gen, 2015; Li & Wu, 2018) reported that the accumulation of effective ingredients in medicinal licorice roots is affected by multiple factors. In this study, the LI content was affected more by the plant species than root depth (Table S3). LI content in Glycyrrhiza uralensis root was significantly higher than in Glycyrrhiza inflata and Glycyrrhiza glabra (Fig. 1A), in line with a previous study (Zhang et al., 2018). We speculate that it might be correlated to the expression of specific functional genes, which might be strongly correlated to the content of effective ingredients, such as glycyrrhizic acid and liquiritin in the root of the licorice species. As per the previous studies (Winkel-Shirley, 2002; Mochida et al., 2017), key functional genes, such as chalcone synthase gene, 3-Hydroxy-3-methylglutary CoA reductase (HMGR), and squalene synthase (SQS), regulate the transcription of glycyrrhizic acid and liquiritin. However, further in-depth analysis is required to characterize the expression of functional genes of effective ingredients. The current study provides a theoretical basis for the developmental strategies related to the improvement of Glycyrrhiza uralensis cultivation. The content of effective ingredients in Glycyrrhiza root samples may result from the interaction between plants and their environment during plant growth. Thus, the accumulation of effective ingredients in root is influenced by its ecological environment. In this study, GIA, GTF, and LI content showed a positive correlation with soil total nitrogen (STN) (r > 0) (Table 1). Thus, we speculate that the majority of the soil nutrients can promote the accumulation of effective ingredients; however, certain soil nutrients, such as soil total potassium (STK), are an exception. Potassium activates multiple enzyme systems and increases stress resistance in plants (Wang & Wu, 2017). In this study, STK was found to be negatively correlated (r < 0) to the GIA, GTF, and LI content, in line with the previous study (Liu et al., 2020). In addition, soil available potassium (SAK) was found to be significantly and positively correlated to GIA level but significantly and negatively correlated to LI level (Table 1). In this study, we speculated that the utilization mechanism of soil nutrients by effective ingredients might be entirely different. However, the underlying mechanism for potassium mediated regulation of effective ingredients remains unclear. Thus, this study may provide platform data for an in-depth analysis. In general, these soil factors exhibit habitat-specific characteristics for regulating the effective ingredients in licorice roots.

In recent years, a growing number of studies (Stegen et al., 2012; Edwards et al., 2015; Nuccio et al, 2016) reported that the dynamics of the microflora is driven to a large extent by environmental factors, such as soil characteristics (pH, nitrogen, phosphorus, and potassium) and climatic conditions (rainfall and temperature). In line with these reports, our study showed that LI, RWC, SAN, and SAK content were the major contributing factors to the variations in the overall structure of the endophytic fungal community (Fig. 6 and Table S7). In addition, we found that the LI content in Glyrrhiza root was significantly and positively correlated to the diversity of endophytic fungal community (Shannon and Simpson index) (P < 0.05) (Fig. 6). Liquiritin (LI), an essential component of flavonoids, confers clinical efficacy to the medicinal licorices and serves as an important quality index for determining the quality of medicinal licorices. Flavonoids synthesis in host plants is induced when the symbiotic fungus is acted upon by purified signaling molecules secreted from the same fungal cells during colonization. Chen et al. (2017) demonstrated that stem biomass, root biomass, and liquiritin content in the root of host plants increased significantly when inoculated with fungi Glomus mosseae, Glycyrrhiza uralensis.

Meanwhile, our results showed that physicochemical factors of the soil and effective ingredients had a significant effect on the composition of endophytic fungal communities. It demonstrated the interaction between endophytic fungal community, root factors, and soil factors. Thus, it indicated that fungal composition could be altered by altering soil factors (Liu et al., 2018), promoting the accumulation of effective ingredients in plants (Han et al., 2013). Xie et al. (2019) showed that in medicinal licorice, P addition and arbuscular mycorrhizal (AM) inoculation improved plant growth and facilitated glycyrrhizic acid and liquiritin accumulation in Glycyrrhiza uralensis. Meanwhile, Orujei, Shabani & Sharifi-Tehrani (2013) also showed arbuscular mycorrhizal fungi (AMF) inoculation enhanced the growth rate and accumulation of effective ingredients in licorice roots (Glycyrrhiza glabra) as compared to control.

In general, the current study unraveled the ecological role of non-biological factors (soil and root) in the endophytic fungal community composition of medicinal licorices. Besides, this study provides crucial information for the developmental strategies related to improving the production and quality of medicinal licorice plants. However, further in-depth studies are required to characterize the functions of the endophytic fungi.

Supplemental Information

Supplemental Information 1 Distribution of the number of tags on each classification level (Kingdom, Phylum, Class, Order, Family, Genus and Species) (a); Rarefaction curves of fungal community composition (b)

Sample Name: Gi, Gg and Gu: Glycyrrhiza inflata, Glycyrrhiza glabra and Glycyrrhiza uralensis; 1, 2 and 3: root depth 0-20 cm, 20-40 cm, and 40-60 cm, respectively; the third number representing the replicate number. The rarefaction curves different colors represent different samples.

Click here for additional data file.

Supplemental Information 2 Geographical sources and physicochemical properties of soil samples

Click here for additional data file.

Supplemental Information 3 The raw data of effective ingredients and physicochemical properties of samples

Raw data of the effective ingredients and physicochemical properties of samples, Gi, Gg and Gu:Glycyrrhiza inflata,Glycyrrhiza glabra and Glycyrrhiza uralensis; 1, 2 and 3: root depth 0–20 cm, 20–40 cm, and 40–60 cm. Abbreviations: GlA, glycyrrhizic acid; GTF, total flavonoid; LI, liquiritin; SOM, soil organic matter; STN, soil total nitrogen; STP, soil total phosphorus; STK, soil total potassium; SNN, soil nitrate nitrogen; SAN, soil ammonium nitrogen; SAP, soil available phosphorus; SAK, soil available potassium; TS, total salt; PH, soil pH; SWC, soil water content.

Click here for additional data file.

Supplemental Information 4 Effect of plant species and root depth on the bioactive compounds of licorice root

P < 0.05 indicates statistical significance.

Click here for additional data file.

Supplemental Information 5 Sequencing results of each sample

Raw reads refers to the sequence filtering out low-quality bases; clean reads refers to the sequence finally used for subsequent analysis after filtering chimeras; base refers to the number of bases of final clean reads; Avglen refers to the average length of clean reads. Q20 refers to the percentage of bases whose quality value is greater than 20 (sequencing error rate is less than 1%); GC (%) refers to the content of GC bases in clean reads; effective (%) refers to the percentage of the number of clean reads and the number of raw reads. Sample name: Gu, Gg and Gi:Glycyrrhiza uralensis,Glycyrrhiza glabra and Glycyrrhiza inflata, respectively; the second number representing root depth 1, 2 and 3: 0–20 cm, 20–40 cm, and 40–60 cm, respectively; the third number representing the replicate number.

Click here for additional data file.

Supplemental Information 6 The alpha diversity indexes in each group

Community richness was identified using the Chao1 and ACE estimator. Community diversity was identified using the Shannon and Simpson indexes. Sequencing depth was characterized by Good’s coverage, good’s coverage estimator values was 99.9%, indicating that the sequence numbers per sample were high enough and has met the requirements. Sample name: Gi, Gg and Gu:Glycyrrhiza inflata,Glycyrrhiza glabra and Glycyrrhiza uralensis; 1, 2 and 3: root depth 0–20 cm, 20–40 cm, and 40–60 cm.

Click here for additional data file.

Supplemental Information 7 Composition of dominant fungi at each classification level

Others: The sum of the undefined and unannotated parts.

Click here for additional data file.

Supplemental Information 8 Results for db-RDA testing effects of soil physicochemical properties and bioactive compounds on the composition and distribution of fungal community in licorice root

r2 is the determinant coefficients of the distribution of the fungal community by environmental factors. Abbreviations: GlA, glycyrrhizic acid; GTF, total flavonoid; LI, liquiritin; SOM, soil organic matter; STN, soil total nitrogen; STP, soil total phosphorus; STK, soil total potassium; SNN, soil nitrate nitrogen; SAN, soil ammonium nitrogen; SAP, soil available phosphorus; SAK, soil available potassium; TS, total salt; PH, soil pH; SWC, soil water content.

Click here for additional data file.

In this study, we would like to thank professor L.Z. for hers guidance, all the authors for their joint efforts. We also would like to thank many graduate students and staff who directed the collection of soil samples that were not listed as co-authors.

Additional Information and Declarations

Competing Interests

Author Contributions

Data Availability

The authors declare there are no competing interests.

Hanli Dang and Tao Zhang conceived and designed the experiments, performed the experiments, analyzed the data, authored or reviewed drafts of the paper, and approved the final draft.

Zhongke Wang conceived and designed the experiments, analyzed the data, prepared figures and/or tables, and approved the final draft.

Guifang Li performed the experiments, prepared figures and/or tables, and approved the final draft.

Wenqin Zhao performed the experiments, analyzed the data, prepared figures and/or tables, and approved the final draft.

Xinhua Lv analyzed the data, authored or reviewed drafts of the paper, and approved the final draft.

Li Zhuang conceived and designed the experiments, authored or reviewed drafts of the paper, and approved the final draft.

The following information was supplied regarding data availability:

The amplicon sequencing reads are available at the NCBI Sequence Read Archive (SRA): PRJNA664554.

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
