# Peer review of "Differences in the endophytic fungal community and effective ingredients in root of three Glycyrrhiza species in Xinjiang, China"

_PeerJ, doi:10.7717/peerj.11047_

## Round 0.1 · original submission · Major Revisions

Dear Authors,

Although the two Reviewers have recognized that your manuscript has merits, they point out a number of critical flaws in the study performed. I agree with their main concerns, which are related to statistical analysis and speculation on results interpretation, being in some situations the conclusions reached by the authors not supported by the data obtained. Therefore, I would be willing to consider a revised version if you are willing to address the comments provided by the two Reviewers.

·

Basic reporting

The article meet the standards of Peer J. However, revision is needed for improvements

Experimental design

Statistical analysis needs authors attention
Data shown in Figure 1 must be analyzed through one way ANOVA and authors need to mention which posthoc test was applied
Correlation study is misleading. Zero means no correlation

Validity of the findings

Findings are valid

Additional comments

The MS reports the effect of root depth and plant species on fungal diversity and root flavonoids, glycyrrhizic acid and liquiritin. It was found that liquiritin content is affected by plant species but not by root depth. However, fungal diversity varied with root depth and plant species.
There are several issues in the study which need author’s attention
1. The MS is written in poor English and need substantial improvement
2. How can total flavonoids be quantified by HPLC as mentioned in abstract?
3. liquiritin contents were not affected by root depth, correlation showed that differences in fungal diversity were due to liquititin. Interestingly, root depth had effect on endophytes diversity. How can authors justify this?
4. In tabe 1, Figure 5 and 6 abbreviations are not explained
5. In Figure 1, DATA must be analyzed by ne way ANOVA? It is not clear which posthoc test is applied?
6. L53-55: How does genetic background of other plants affect bioactive compounds of a medicinal plant?
7. Methods section contains some unnecessary details such as L116-121. Delete such details
8. In method section mention Detector, mobile phase and Elution rate of HPLC
9. L214-216: what is meant by W, D and G? explain on first appearance
10. L220 and elsewhere: 0 means no correlation at all?
11. Discussion needs improvement. Include logical reasoning i.e. why fungal diversity varies with root depth? What is the possible reason of ascomycetes abundance ect.
12. conclusion is a mere repetition of results.

Reviewer 2 ·

Basic reporting

1.1. Figures are relevant, but show various styles, they seem to come from various software (Fig. 1. and 3a show a different style vs the others). Please address. Additionally, taxonomic names should be italic in captions and statistical tests should also be included.

1.2. English needs minor corrections, but otherwise the paper is clearly written. Please correct this non-exhaustive list of errors:
L44: new sentence, ". Its dried roots"
L46: "Glycyrrhizic acid, the richest content of triterpene saponins [5]," - perhaps you meant "Glycyrrhizic acid, the chief triterpene saponin"
L65: there is no need to write hormone names with capital letters
L75: typo "officinale"
L82: "on routinely used media"
L382: can regulate

1.3. I do not understand why G. uralensis is abbreviated "W", G. inflata as "D" when G. glabra is "G".

1.4. Introduction is good. I'd add that a few relevant literature remained uncited: 10.1016/j.cpb.2020.100154, 10.21276/ap.2016.5.2.19. As no metagenomics is done in these, novelty is not compromised.

1.5. Structure conforms to PeerJ standards, but the applied reference format is not author - date format.

1.6. Raw data were shared.

Experimental design

2.1. The applied methods confirm accepted methodologies, with the exception of no p-value threshold adjustment for doing a high number of statistical tests. This is a major issue. Such a high amount of correlation tests requires the adjustment of p-value thresholds to set the hypothesis (or better, study-) level error chance to 0.05. See 10.1007/s11306-006-0037-z. This adjustment will tag many of your current results as false-positives. To cope with the issue, the amount of statistical tests can be reduced by subjecting the dataset to statistical tests after reduction of dimensionality.

2.2. I'd speak of correlations above an absolute value of 0.6 - 0.7. With n = 3 you simply do not have the statistical power to test these relationships successfully. The low values in Table S7 and a fault in experimental design makes the claims in L300-316 unsupported in my opinion. A more prudent discussion of the results is warranted.

2.3. Minor notes:
L177: Please state what was the amount of sequences of plant origin in your raw dataset.
L196: Why is such an old R version used? Did you accidently write the version of a used R package?

2.4. An original primary research within Scope of the journal. No ethical issues. Methods are descrbied in sufficient detail. Research question is well defined and relevant. There is not much information available on the microbiome of this important medicinal plant. I think the investigation of the depth gradient is especially interesting.

Validity of the findings

3.1. The biggest problem with the experimental design is that each soil type was represented by a single species only that limits the possible conclusions that can be drawn from the findings. The authors are fortunately aware of this limitation, as written in L387-391. But this, unfortunately this prevents drawing conclusions about the effects of the soil type and the plant species, as the interaction term cannot be estimated from these data. The situation is worsened by the low statistical power (n=3). Perhaps this is why we see extreme within-sample variability, as presented in the db-RDA plot in Figure 7. This also questions the usefulness of Table 1. I understand that the answers can be answered only by introducing all test species to all sampling sites, which is not realistic. Therefore, I suggest conversion of the claims regaring the effects of soil parameters and species to speculative sentences, which is accepted by the journal. The undoubtedly well-supported data are limited to the fungal species list in each Glycyrrhiza, and the depth-gradient values. Please revise the study accordingly.

---

## Round 0.2 · Minor Revisions

Dear Authors,

I appreciate all your substantial efforts to improve the paper according to the reviewer's comments and suggestions. In this form, the manuscript has improved considerably. Still, the reviewers raised substantial concerns about statistical analysis. Thus, before proceeding with my decision I would like to ask you to address the comments provided by the two reviewers, in particular reviewer 2, in your revision.

·

Basic reporting

The article is substantially improved and is suitable for publication now

Experimental design

Good

Validity of the findings

Results are valid

Additional comments

Please give attention to my previous comment no. 3.
In abstract you mentioned that root depth had no effect on liquititin contents, in reply to comment you pointed that root depth had significant influence on liquititin.

Reviewer 2 ·

Basic reporting

1.1. Fig.1. caption shows letters for the post-hoc test but the figure has '*' in the caption. Please address.

1.2. Abbreviating G. uralensis as "W", G. inflata as "D" and G. glabra as "G" is (still) disturbing. In my previous point, I did not argue against using abbreviations, I just meant why not use Gu, Gi, Gg, or something similar? One always has to check again which letter is which species.

Experimental design

2.1. The 2.1. point of my previous comments was not adequately addressed. In particular, I still see no p-value threshold adjustment for doing a high number of statistical tests - Fig. 5. is the same as before, the authors apparently misunderstood my comment, but this is an important issue. Such a high amount of correlation tests requires the adjustment of p-value thresholds to set the hypothesis (or better, study-) level error chance to 0.05. Again, see 10.1007/s11306-006-0037-z.

In particular, in Fig. 5., and in section L296-313, p < 9.52381e-05 would be required to pass a significance test if Bonferroni correction was truly applied (0.05 / n, where n is the number of statistical tests). Logically, this adjustment will tag many of your current positive results as false-positives. Doing hundreds of tests logically results in some p<0.05 results.

To cope with the issue, the amount of statistical tests can be reduced by subjecting the dataset to statistical tests after reduction of dimensionality (e.g. grouping multicorrelating groups together). Anyway, if such high amounts of statistical tests are done, you should not claim high significane for phenomena with p ~ 0.01. Also see 3.1.

The most sound solution would be to accept that you did not have the experimental setup and/or sample size to study soil parameter - microbial composition correlations that lead to statistically significant results - you still have one species per site. This would result in removing results in L296-313, but this would not reduce the value of your work as you perhaps think.

2.2. Response to previous 2.2.
Statistical significance of a correlation coefficient means you have enough data and good signal-to-noise ratio (hence, statistical power) to be able to say the correlation is truly there, it's "real". This does not make a correlation "strong" or "weak". Therefore, phrases such as "highly significant but negative correlation" (L300) are perhaps signs of misunderstanding. Don't get me wrong, I am not expecting 0.99 R2 values from a biological experiment. But you should ONLY discuss correlations that passed the statistical tests, the rest should be considered noise. I doubt you have enough data to support the claims in L296-313. Please give four-digit R2 values throughout the results section instead of writing "r > 0".

Validity of the findings

3.1. The Discussion section seems to be cautious enough in its current form.

Additional comments

None.

---

## Round 0.3 · Minor Revisions

Dear Authors,

Thank you for addressing all the comments of the reviewers. Before proceeding with my decision I would like to ask you to please consider the following revisions:

1) In figure 6, please replace "W", "D" and "G" by the new abbreviations Gi, Gg and Gu to indicate Glycyrrhiza inflata, Glycyrrhiza glabra and Glycyrrhiza uralensis. These abbreviations should also be indicated in the figure legend.
2) In figure 4, please check the abbreviations (G)
3) Check the references in line 327 and line 585

---

## Round 0.4 · accepted · Accept

Dear Dr. Zhuang,

Thank you for addressing all the comments.

I am pleased to inform you that your work has now been accepted for publication in PeerJ. Congratulations on the excellent work!

With kind regards,
Paula Baptista